

# Producing reliable hydrologic scenarios from raw climate model outputs without resorting to meteorological observations

Simon Ricard[1,2], Philippe Lucas-Picher[3,4], Antoine Thiboult[2], and François Anctil[2]

[1]Institut de recherche et de développement en agroenvironnement (IRDA), Québec, Canada
5   [2]Département de génie civil et de génie des eaux, Université Laval, Québec,
[3]Groupe de Météorologie de Grande Échelle et Climat (GMGEC), Centre National de Recherches Météorologiques (CNRM),
Université de Toulouse, Météo-France, Centre National de la Recherche Scientifique (CNRS), Toulouse, France
[4]Département des sciences de la Terre et de l'atmosphère, Université du Québec à Montréal, Montréal, Québec, Canada
Canada

10   *Correspondence to*: Simon Ricard (simon.ricard@irda.qc.ca)

**Abstract.** A simplified hydroclimatic modelling workflow is proposed to quantify the impact of climate change on water discharge without resorting to meteorological observations. This alternative approach is designed by combining asynchronous hydroclimatic modelling and quantile perturbation applied to streamflow observations. Calibration is run by forcing hydrologic models with raw climate model outputs using an objective function that exclude the day-to-day temporal correlation between simulated and observed hydrographs. The resulting hydrologic scenarios provide useful and reliable information considering: (1) they preserve trends and physical consistency between simulated climate variables, (2) are implemented from a modelling cascade despite observation scarcity, and (3) support the participation of end-users in producing and interpreting climate change impacts on water resources. The proposed modelling workflow is implemented over four subcatchments of the Chaudière River, Canada, using 9 North American CORDEX simulations and a pool of lumped conceptual hydrologic models. Results confirm that the proposed workflow produces equivalent projections of the seasonal mean flows in comparison to a conventional hydroclimatic modelling approach. They also highlight the sensibility of the proposed workflow to strong biases affecting raw climate model outputs, frequently causing outlying projections of the hydrologic regime. Inappropriate forcing climate simulations were however successfully identified (and excluded) using the performance of the simulated hydrologic response as a ranking criterion. Results finally suggest further works should be conducted to confirm the reliability of the proposed workflow to assess the impact of climate change on extreme hydrologic events.



## 1 Introduction

Assessments of climate change impacts are commonly oriented in a top-down perspective favouring the implementation of a modelling cascade from greenhouse gas concentrations to hydrologic (impact) models (e.g. Poulin et al., 2011; Seiller and Anctil, 2014; Seo et al., 2016). Since climate models are affected by uncertainties that limit their ability to simulate atmospheric processes at the local scale, statistical post-processing is typically applied to bias correct their (raw) outputs to improve agreement with in-situ observations. The product of a post-processed climate simulation is often termed climate scenario: a plausible trajectory that originally shares the statistical properties of the local (reference) recent past and evolves along physically based long-term trends (Mearns et al., 2001; Huard et al., 2014). Resulting climate scenarios are subsequently translated into simulated streamflow series using calibrated hydrologic models.

Usage of post-processed climate model outputs is criticized for three main reasons (e.g. Alfieri et al., 2015b, Chen et al., 2018; Lee et al., 2018): (1) it disrupts the physical consistency between simulated climate variables; (2) it affects the trend in climate change signals imbedded within raw climate simulations; (3) it requires abundant good-quality meteorological observations, which are unavailable for many regions of the world, including some less common meteorological fields such as wind speed, relative humidity, and radiation (Ricard et al., 2020). More marginal critics raise the fact that statistical post-processing hides raw climate model outputs biases from end-users (Ehret et al., 2012), potentially blurring confidence attributed to resulting impact scenarios and misleading adaptation to climate change. Even if these limitations are generally acknowledged, statistical post-processing is often considered mandatory to climate change impact assessment studies on water resources. Trend-preserving and multivariate approaches (e.g. Cannon et al., 2018; Ahn and Kim, 2019; Nguyen et al., 2020) have been specifically developed in order to limit the above-mentioned post-processing drawbacks. However, these approaches involve a fairly high level of complexity and, consequently, requires specific expertise in post-processing technologies.

In the scientific literature, raw climate model outputs are mostly used as benchmarks to assess the performance issued by post-processed climate model outputs (e.g. Teng et al., 2015; Ficklin et al., 2016; Charles et al., 2020). The use of raw climate model outputs as hydrologic scenarios is a marginal practice, mostly because resulting streamflow simulations are correspondingly affected with biases (e.g. Muerth et al., 2013) and by the lack of synchronicity between the simulated climate and the observed hydrologic (river discharge) time series. Such implementation is mostly justified when focusing on relative changes to reference conditions (Alfieri et al., 2015a,b) or under the assumption that climate model output biases are sufficiently small to be compensated by the calibration of the hydrologic model (Chen et al., 2013). It is also justified when extreme events are analyzed considering the uncertainty introduced by the short sampling of observation chronicles (Meresa and Romanowicz, 2017). Advocating the benefit of preserving the dependence between simulated climate variables, Chen et al. (2021) recently constructed hydrologic scenarios from raw climate model outputs by applying the daily-translation bias correction method (Mpelasoka and Chiew, 2009) to streamflow simulations instead to climate simulations. The authors demonstrated the approach reduces hydrologic biases comparably to a conventional one for which climate simulations are



corrected beforehand. They finally highlighted that, regardless of the modelling approach, climate simulations issue poor
hydrologic responses due to the non-stationarity of the climate biases and abrupt seasonal fluctuations affecting correction
factors.

Most climate change studies resort to a modelling cascade for which the hydrologic model is calibrated independently of the
climate model outputs, using observations as meteorological forcings (e.g. Poulin et al., 2011; Seiller and Anctil, 2014; Seo et
al., 2016). This is questionable since calibration then compensates errors from meteorological observations (e.g. solid
precipitation undercatch or spatial interpolation of in-situ observations) but not to those from climate models outputs. It
consequently influences the identification of hydrologic model parameters, as well as the representation of hydrologic
processes simulated at the catchment scale. The resulting effect on the hydrologic scenarios and projected changes of the water
regime components remains mostly misunderstood. Few studies conducted calibration by forcing hydrologic models directly
with raw climate model outputs. Chen et al. (2017) quantified the hydrological impacts of climate change over North America,
calibrating a lumped conceptual hydrologic model with raw regional climate model (RCM) outputs over a recent past period.
Ricard et al. (2019) proposed an alternative configuration of the hydroclimatic modelling chain and tested five objective
functions that exclude the temporal synchronicity of hydrologic events, such as the correspondence between observed and
simulated targeted quantiles, distribution moments, mean flows, or annual cycles. They concluded that forcing a physically-
based hydrologic model with regional climate simulations according to asynchronous modelling principles can improve the
simulated hydrologic response over the historical period. Ricard et al. (2020) implemented statistical post-processing of raw
climate model outputs within the asynchronous modelling framework by calibrating quantile mapping transfer functions
together with the parameters of the hydrologic model. They integrated relative humidity, solar radiation and wind speed, for
which observations are scarce or unavailable, to a modelling chain and confirmed the improvement of the simulated hydrologic
response in comparison to a conventional framework using reanalyses as a description of the reference climate.

This study proposes a straightforward hydroclimatic modelling workflow enabling the production of streamflow projections
without post-processing climate model outputs and without using meteorological observations. The procedure is inline with
the modelling frameworks experimented by Ricard et al. (2020) and Chen et al. (2021). In essence, the workflow translates
raw climate model outputs into a corresponding simulated hydrologic response using an asynchronous framework that encrypts
simulated hydrologic changes by defining change factors for each streamflow quantiles. When relative trends are required, a
qualitative climate change impact assessment can be conducted by analysing the distributions of change factors. When
hydrologic time series are required, the change factors can be applied on the available streamflow observations. This approach,
referred to as quantile perturbation, has been previously applied to climate model outputs (e.g. Sunyer et al., 2014; Willems
and Vrac, 2011) but not, to our knowledge, using streamflow simulations resulting from a hydroclimatic modelling cascade.
The key advantage of the proposed approach is that meteorological observations are not required, nor for post-processing
climate model outputs, nor for calibrating the hydrologic model. It is thus easy to implement compared to the conventional





modelling cascades, which are typically affected by much heavier requirements in terms of data, modelling processes, and computing capacity. The workflow also preserves trends and consistency between simulated climate variables and allows a bottom-up assessment of raw climate model outputs from the perspective of the impact modeler and end-users expertise. The study also aims to assess and discuss its reliability by comparison to a conventional hydroclimatic modelling, involving post-

processing of raw climate model outputs and calibration of hydrologic models using meteorological observations. Section 2 presents the watershed of interest of the data used in the study. Section 3 explains the methodological specificities of the proposed workflow and describes its implementation using raw NA-CORDEX simulations over a mid-scale catchment located in Southern Québec, Canada, and a pool of lump conceptual hydrologic models. Section 4 displays results while Section 5 discusses strengths and weaknesses of the proposed modelling workflow.

## 2 Watershed of interest and data

The study is conducted over four subcatchments of the Chaudière River (Fig. 1), a 185-km river that takes its source in Mégantic Lake (altitude 395 m) and flows northward into the Saint-Lawrence River, near Québec City. The 6694-km² catchment is located in the southern part of Province of Québec, Canada, bordering the United States at its meridional delineation. It is shaped by a moderate topography (the highest peak is 1100 m) mostly corresponding to the Appalachian

geological formation upstream and the Saint-Lawrence Lowlands downstream. The river slope is steep (~2.5 m/km) upstream of the town of Saint-Georges (site 4, Fig. 1) and abruptly gentles to ~0.5 m/km down to Saint-Lambert (site 2). The catchment is mostly covered by forest (~70 %), agricultural land uses are nonetheless substantial (~23 %), mostly in the lower portion of the catchment. The Chaudière River frequently floods from Saint-Georges down to Saint-Lambert and is also prone to ice jams mostly around Beauceville (roughly 10 km downstream of Saint-Georges).

The climate is humid continental (Dfb according to the Köppen classification). The mean annual temperature shows marked seasonal fluctuations (see Fig. 3), falling below freezing roughly from November to March. Total annual precipitation is around 1000 mm, depicting no seasonal fluctuations except for a mild intensification from August to November. The corresponding hydrologic regime can be categorized as nivopluvial, corresponding to an alternance of two dominant flood periods. Driven by snowmelt and rainfall, the main flood period takes place from March to April, while the secondary in autumn is driven by

an increase of precipitation. These two flood-prone periods are punctuated by two low flow periods. The flow regime is mostly free from the influence of dam operation, except for short river reaches downstream of Mégantic and Sartigan dams (located at Mégantic Lake and upstream from Saint-Georges, respectively).

Nine North American (NA-) CORDEX simulations (Mearns et al., 2017; Table 1) are used to construct the hydrologic scenarios. They consist of 50-km Regional Climate Models (RCM) simulations that are driven by four Global Climate Models

(GCM) forced by the RCP8.5 greenhouse gas (GHG) concentrations. Daily minimum and maximum 2-meter air temperature and daily precipitation were archived over a reference historical period from 1970 to 1999 and a future period from 2040 to





2069. Since no statistical post-processing is applied in the proposed modelling workflow, RCM simulations are preferred to GCM simulations to minimize the scale mismatch between the climate models and the in-situ observations. RCP8.5 is preferred over RCP4.5 for its more pronounced climate change signal and because more NA-CORDEX simulations are then available.

Since the studied catchment features a topography of moderate complexity and a medium area of 6694 km², a 50-km horizontal resolution was considered sufficient over the finer, but smaller ensemble of 25-km simulations. Other climate change impact studies have relied on a comparable number of RCM simulations (e.g. Alfieri et al., 2015a,b; Laux et al., 2021).

Daily discharge observations are collected from the Québec hydrometric network (MELCC, 2021). Stations located at the outlets of the four subcatchments of the Chaudière River are described in Table 2. The four subcatchments encompass 87% of

the area of the Chaudière River catchment, only the very downstream part is ungauged. Hydrometric stations 023402 (site 2) and 023429 (site 4) are located on the main river, while 023401 (site 1) and 023422 (site 3) are located on the Beaurivage and Famine rivers, two important effluents (709 and 691 km², respectively). Streamflow observational record lengths are fairly long according to North American standards. 023401 and 023402 are in operation since early 20th Century, while 023422 and 023429, from 1964 and 1969, respectively. The gridded observation datasets (daily air temperature and precipitations) are

derived from kriging in situ data at 0.1° resolution from 1970 to 2018 (Bergeron, 2015). For the study, we extracted the observed time series from 1970 to 1999.

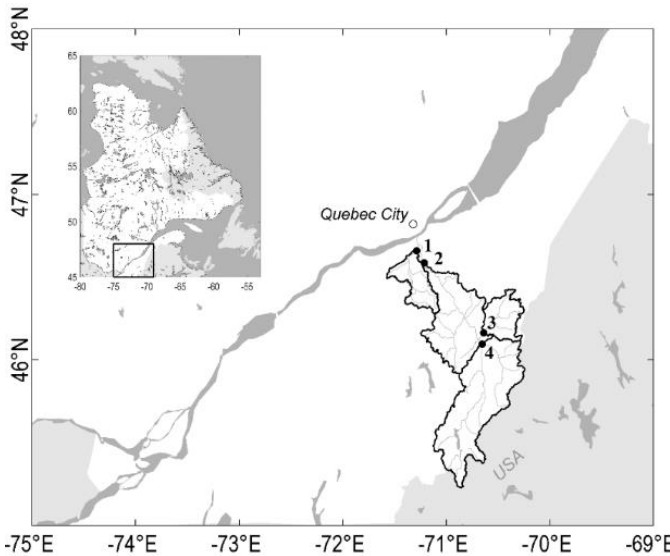

**Figure 1: Location of the Chaudière River and subcatchments described in Table 2. Sites 1 to 4 correspond to the location of hydrometric stations.**



**140**    **Table 1. Description of North American CORDEX simulations**

| ID | GCM | RCM | Resolution | RCP | Reference period | Future period |
|----|-----|-----|-----------|-----|------------------|---------------|
| crx1 | CanESM2 | CRCM5 | | | | |
| crx2 | CanESM2 | CanRCM4 | | | | |
| crx3 | CanESM2 | RCA4 | | | | |
| crx4 | EC-EARTH | HIRHAM5 | | | | |
| crx5 | EC-EARTH | RCA4 | 50 km | 8.5 | 1970-1999 | 2040-2069 |
| crx6 | GFDL-ESM2M | RegCM4 | | | | |
| crx7 | MPI-ESM-LR | CRCM5-UQAM | | | | |
| crx8 | MPI-ESM-LR | RegCM4 | | | | |
| crx9 | MPI-ESM-LR | WRF | | | | |

**Table 2. Description of Chaudière River subcatchments**

| Site | Hydrometric station ID* | Location | River | Area (km²) | Data availability |
|------|-------------------------|----------|-------|-----------|-------------------|
| 1 | 023401 | Lévis | Beaurivage | 709 | 1925-today |
| 2 | 023402 | Saint-Lambert | Chaudière | 5820 | 1915-today |
| 3 | 023422 | Saint-Georges | Famine | 691 | 1964-today |
| 4 | 023429 | Saint-Georges | Chaudière | 3070 | 1969-today |

*Notification attributed by the Québec hydrometric network (MELCC, 2021).*

## 3 Methods

### 3.1 The proposed modelling workflow

**145**    The proposed asynchronous modelling workflow (Fig. 2) aims to construct climate change impact hydrologic scenarios without resorting to meteorological observations, nor for post-processing climate model outputs, nor for calibrating hydrologic models. It follows three main steps: (1) translating raw climate model outputs into corresponding hydrologic responses using asynchronous modelling, (2) computing change factors derived from a reference and a future simulated hydrologic response, and (3) constructing hydrologic scenarios by applying correction factors to the available streamflow observations.

**150**    Asynchronous hydroclimatic modelling (Ricard et al., 2020) refers to an alternative configuration of the hydroclimatic modelling chain for which the calibration is performed on climate model outputs (over a recent past reference period) and not on meteorological observations. Since climate models cannot reproduce the observed sequence of meteorological events, the parameters of hydrologic models are optimized according to an objective function that purposely exclude the day-to-day temporal correlation (Ricard et al., 2019).



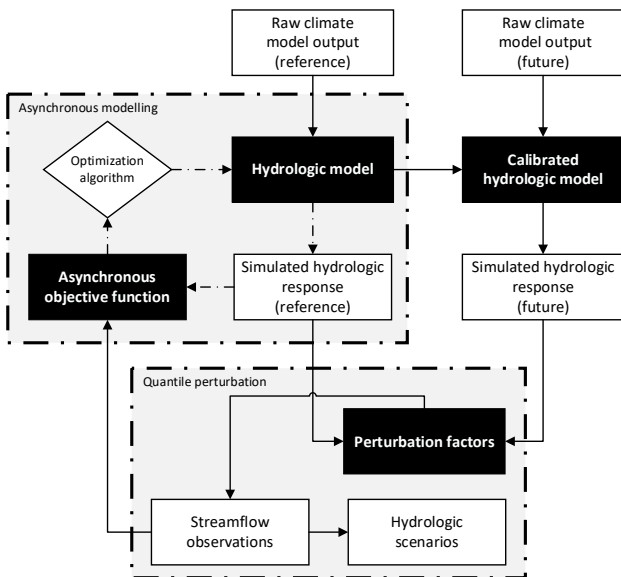


**Figure 2: The proposed asynchronous modelling workflow. In comparison to a conventional hydroclimatic modelling approach, the production of hydrologic scenarios does not require meteorological observations, nor for post-processing raw climate modelling, nor for calibrating the hydrologic model.**

The calibration loop trains the hydrologic model in reproducing the statistical properties of the streamflow regime such as the

form of its cumulative distribution, quantiles, or moments, without taking into account the temporal match between them. We propose here a normalized score inspired by the CRPS (Matheson and Winkler, 1976) where the distribution of simulated streamflow is compared against the distribution of observations – the CRPS is commonly used to assess ensemble prediction systems. More specifically, the proposed score is defined such that:

$$nCRPS(F, x_{obs}) = \int_{-\infty}^{\infty} \big(F(\tilde{x}) - F(\tilde{x}_{obs})\big)^2 \, dx \qquad (1)$$

where $\tilde{x}$ and $\tilde{x}_{obs}$ are respectively the normalized simulated and observed streamflow time series, and $F(\tilde{x})$ refers to the temporal cumulative distribution of the streamflow.

$$\tilde{x} = \frac{x}{\max(\{x, x_{obs}\})} \qquad (2)$$

$$\tilde{x}_{obs} = \frac{x_{obs}}{\max(\{x, x_{obs}\})} \qquad (3)$$

In simple terms, the nCRPS is the squared difference between the normalized observed and simulated cumulative distribution

functions, integrated with respect to the normalized streamflow. A perfect similarity between the two distributions indicates that the simulated values share the same statistical properties than the observations. In such case, the area between the two curves would be null and the nCRPS equals to 0. The calibration loop being completed, raw climate model outputs are translated into corresponding hydrologic responses by forcing the calibrated hydrologic model over an application period, typically including both reference and future ones.



Hydrologic scenarios are constructed by applying a non-parametric quantile perturbation (Willems and Vrac, 2011) to the streamflow observations. Assuming stationarity of climate model biases, quantile perturbation (see also Willems, 2013; Sunyer et al., 2014; Hosseinzadehtalaei et al., 2018) typically modifies meteorological observations according to relative changes in corresponding distributions projected by raw climate model outputs, preserving the simulated meteorological trends in all quantiles, including their tails (Cannon et al., 2015). In the proposed workflow, change factors are defined by relating quantiles

of the simulated reference and future hydrologic responses produced by asynchronous modelling. Change factors ($\phi$) are here defined as the ratio between the simulated streamflow values ($x$, associated to the exceedance probability $p$) of a future ($Fut$) to a reference ($Ref$) period. Change factors encrypt projected trends for each streamflow quantiles such that:

$$\phi(p,t) = \frac{x_{Fut}(p,t)}{x_{Ref}(p,t)} \quad\quad\quad\quad\quad\quad\quad\quad\quad\quad\quad\quad (4)$$

where $t$ refers to a given temporal resolution, i.e. a prior subsampling of the annual cycle for which $\phi$ is evaluated (e.g. bi-

annual, seasonal, monthly).

At this point, the future hydrologic regime can be assessed in terms of relative changes by analyzing change factors for streamflow quantiles of interest. Hydrologic scenarios ($x_{sce}$) are constructed by applying change factors ($\phi$) to the available observed streamflow series ($x_{obs}$) such that:

$$x_{sce}(p,t) = x_{obs} \cdot \phi(p,t) \quad\quad\quad\quad\quad\quad\quad\quad\quad\quad\quad\quad (5)$$

Resulting hydrologic scenarios stand for plausible trajectories of the water regime conditions arising from a given climate simulation ensemble, statistically equivalent to the observed recent past that is affected by a physically based long-term trends.

**3.2 Hydrologic modelling**

Table 3 lists the seven lumped conceptual hydrologic models used for simulating the hydrologic response corresponding to the nine NA-CORDEX simulations. The models are derived from various scientific and operational sources available from the

HOOPLA open source MATLAB® toolbox (Thiboult et al., 2019). Models can be categorized as of moderate complexity, the number of open parameters ranging from 6 to 9. All models are combined to the Oudin evapotranspiration formulation (Oudin et al., 2005) and the snow module developed by Valéry et al. (2014), for which the two parameters, thermal inertia of the snowpack (Ctg = 0.25, adimensional) and a degree-day melting factor (Kf = 3.74 mm d$^{-1}$), are being fixed to default values that are relevant to the region. The selection of hydrologic models is based on the diversity of their structures and their

combined performance for short-term streamflow forecasting (Valdez et al., 2021). The main idea here is to select a pool of heterogenous models in order to avoid that the simulated hydrologic responses are tainted by a single model structure.





**Table 3. Description of the lumped conceptual hydrologic models**

| ID | Inspired from | No. of parameters | No. of reservoirs |
|----|---------------|-------------------|-------------------|
| 1 | CEQUEAU (Girard et al., 1972) | 9 | 2 |
| 2 | HBV (Bergström and Foreman, 1973) | 9 | 3 |
| 3 | IHACRES (Jakeman et al., 1990) | 7 | 3 |
| 4 | MORDOR (Garçon, 1999) | 6 | 4 |
| 5 | PDM (Moore and Clarke, 1981) | 8 | 4 |
| 6 | SACRAMENTO (Burnash et al., 1973) | 9 | 5 |
| 7 | XINANJIANG (Zhao et al., 1980) | 8 | 4 |

Hydrologic models are calibrated according to an asynchronous modelling framework, i.e. being forced with raw climate
model outputs and excluding the day-to-day temporal correlation (Ricard et al., 2019). The calibration loop is run from 1970 to 1979 with the Shuffle Complex Evolution algorithm (Duan et al., 1993) using 10 complexes. A 10-year period is usually considered sufficiently long for calibration, offering a sound trade-off between identifying representative parametric values and computational requirements.

### 3.3 Conventional hydroclimatic modelling

The proposed asynchronous workflow is compared to a conventional top-down hydroclimatic modelling approach. The latter is typically implemented to produce hydrologic scenarios from GCM or RCM simulations following three main steps (e.g. Poulin et al., 2011; Seiller and Anctil., 2014; Seo et al., 2016). Raw climate model outputs are first being post-processed to correct systematic errors (or biases) according to available meteorological observations or reference product describing the climate system over a recent and sufficiently long past period. Simulated 2m minimum and maximum air temperature and
precipitation are corrected using a quantile mapping approach (Lucas-Picher et al., 2021) combined to daily local intensity scaling (Schmidli et al. 2006). Quantile mapping is implemented every month using 100-nodes transfer-functions interpolated linearly. The wet day frequency is corrected using a 0.1-mm threshold. Hydrologic models are calibrated separately, forced with gridded meteorological observation datasets to optimize the performance of the simulated hydrologic response according to available streamflow observations. Hydrologic scenarios are finally constructed by forcing the calibrated hydrologic models
with post-processed climate model outputs. For the sake of comparison, hydrologic modelling within the conventional hydroclimatic approach is implemented equivalently to the asynchronous workflow as described in Sections 3.2, using the same pool of hydrologic models, the same objective function, calibration period, and configuration of the optimization algorithm.



## 4 Results

4.1 Biases and projected changes of NA-CORDEX 2-meter air temperature and precipitation

Figure 3 illustrates the annual cycle of the 2-meter mean air temperature (2mt) simulated by the nine NA-CORDEX simulations from 1970 to 1999. Only subcatchment 2 is shown considering it represents most of the area of the Chaudière River catchment, but also because subcatchments 3 and 4 are nested within. Corresponding observations issued by interpolation of in-situ measurements and biases are also illustrated. Most climate simulations overestimate 2mt from November to March, the median bias of the ensemble reaching roughly +5 °C in January. NA-CORDEX simulations generally provide a reasonable representation of temperature from May to September, individual biases then ranging from -2 °C to +2 °C from one simulation to another. 2mt biases appear to be linked to the forcing GCM simulations. CanESM2-driven simulations (crx1 to crx 3) lead to similar annual profiles marked with an alternance of high warm winter biases and subsequent moderate warm summer biases. EC-EARTH-driven simulations (crx4 and crx5) show a similar annual profile than CanESM2, but are affected by marked cold spring and summer biases, reaching -5 °C in April in the case of crx4. GFDL-ESM2M-driven simulation (crx6) is affected by a quasi systematic cold bias. MPI-ESM-LR-driven simulations (crx7 to crx9) finally show a constant cold bias from May to November. The winter warm bias carried by crx7 (CRCM5-UQAM, positive) differ however from the winter cold biases of crx8 and crx9 (RegCM4 and WRF).

Figure 4 illustrates the mean annual cycle of the precipitation simulated by the nine NA-CORDEX simulations from 1970 to 1999 over subcatchment 2. The ensemble mean overestimates precipitation by roughly +0.5 mm/day. In opposition to 2mt, biases in precipitation are fairly constant throughout the whole annual cycle, except for a brief period in autumn (August to October) when simulations are less biased. Biases typically range between -1 to +2 mm/day depending on the period of the year. Part of the wet bias in winter precipitation can be explained by solid precipitation undercatch, which can reach 20 to 70 % (Pierre et al., 2019). Also, in opposition to 2mt, biases in annual profiles are not as clearly related to the driving GCM.





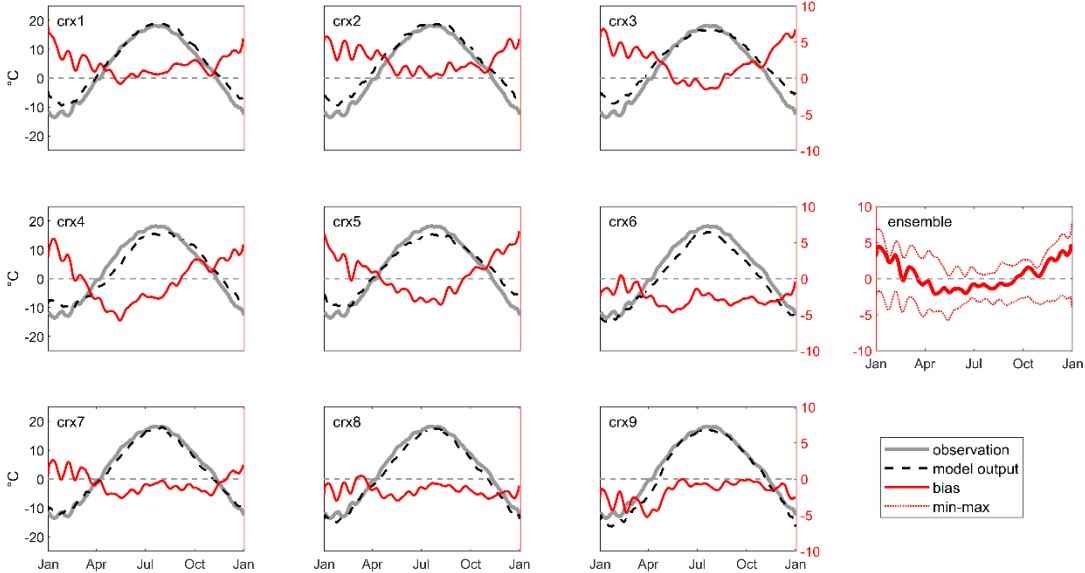

**Figure 3: 2-meter mean air temperature annual cycle simulated by the nine NA-CORDEX simulations (crx1 to crx9) for subcatchment 2, from 1970 to 1999. Observations and biases are presented. The left scale of the y-axis refers to observations and raw climate model outputs, while the right scale, to biases. A 5-day moving window is applied to all time series to enhance the signal to noise ratio. In the ensemble panel, the median, minimum and maximum biases from the nine climate simulations are illustrated. Observations are derived from the kriging of in-situ data.**


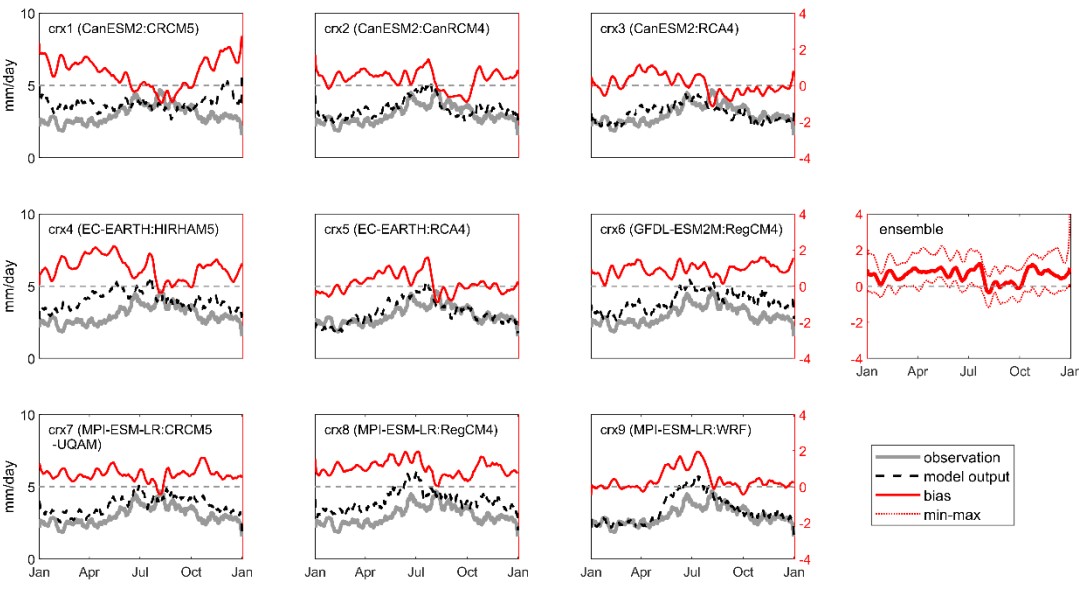

**Figure 4: As Figure 3 but for precipitation.**





Figure 5 illustrates seasonal changes (2040-2069 related to 1970-1999) for subcatchment 2 for the mean 2mt and precipitation from the nine NA-CORDEX simulations. Increase in 2mt generally falls between +2 and +4 °C. Also, most simulations anticipate precipitation to increase in winter (+10 to +25 %), spring (up to +20 %), and autumn (up to +15 %), but to decrease in summer (down to -10 %). Some simulations reveal outlying trends, especially crx3 and crx4 that display, respectively, a +44 % increase in winter precipitation and almost no change in 2mt from September to November.

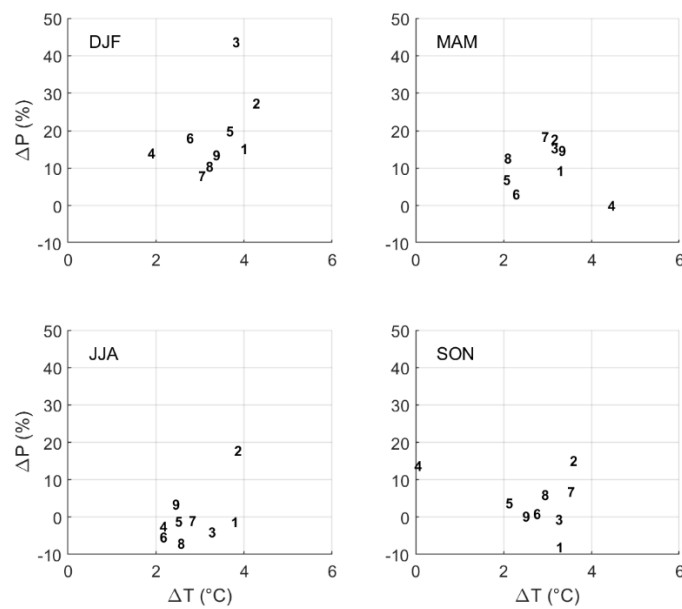


**Figure 5: Projected changes (2040-2069 with respect to 1970-1999) of mean 2mt and precipitation from the nine NA-CORDEX simulations for winter (DJF), spring (MAM), summer (JJA), and autumn (SON) for subcatchment 2. Numbers refers to the crx simulation described in Table 1.**

4.2 Validation of the asynchronous modelling workflow

Figure 6 displays the observed mean annual hydrographs at site 2 over a recent past reference period (1970-1999). The hydrograph shows typical seasonal fluctuations marked by spring flood (~9.5 mm d$^{-1}$) in April and a second peak (much smoother, ~1.8 mm d$^{-1}$) in November. Figure 6 also compares mean annual hydrographs simulated by the asynchronous framework and the conventional hydroclimatic approach for each NA-CORDEX simulation. Results shows the capacity for the conventional approach to provide a more accurate representation of seasonal streamflow fluctuations over the reference

period. Although slightly delayed and underestimated, the peak flow simulated in spring by the conventional approach is typically more accurately synchronized with observations relative to the asynchronous workflow. The inter-model variation (indicated by the envelopes in Figure 6) related to the conventional approach also tends to be smaller and more centered around





the observations, noticeably during summer, fall, and winter. The shape of the simulated hydrographs remains finally quite similar from one NA-CORDEX simulation to another.

Hydrographs simulated by the asynchronous workflow are in some cases affected by notable flaws in representing seasonal streamflow fluctuations. The shape of the simulated hydrographs also differs notably from one NA-CORDEX simulation to another. This can be related to biases affecting raw forcing NA-CORDEX simulations (see Figures 3 and 4). In many cases (crx4, crx6, crx8 and crx9), the spring flood is notably delayed and occurs in late spring. This could be explained by cold biases affecting simulated air temperature in spring, combined in some cases to an overestimation of solid precipitation in winter.

The inter-model variation also tends to be larger relative to the conventional approach, more noticeably during summer and autumn (crx1, crx2, crx3, and crx5), but also in winter (crx1).

Hydrographs simulated at sites 1, 3 and 4 are given in Appendix 1 and lead to equivalent conclusions. Hydrographs simulated by the conventional approach at site 3 however produce an atypical two-fold spring flood that can be related to a specific hydrologic model. The inter-model variability is also more marked in the case of site 4 for the asynchronous framework.

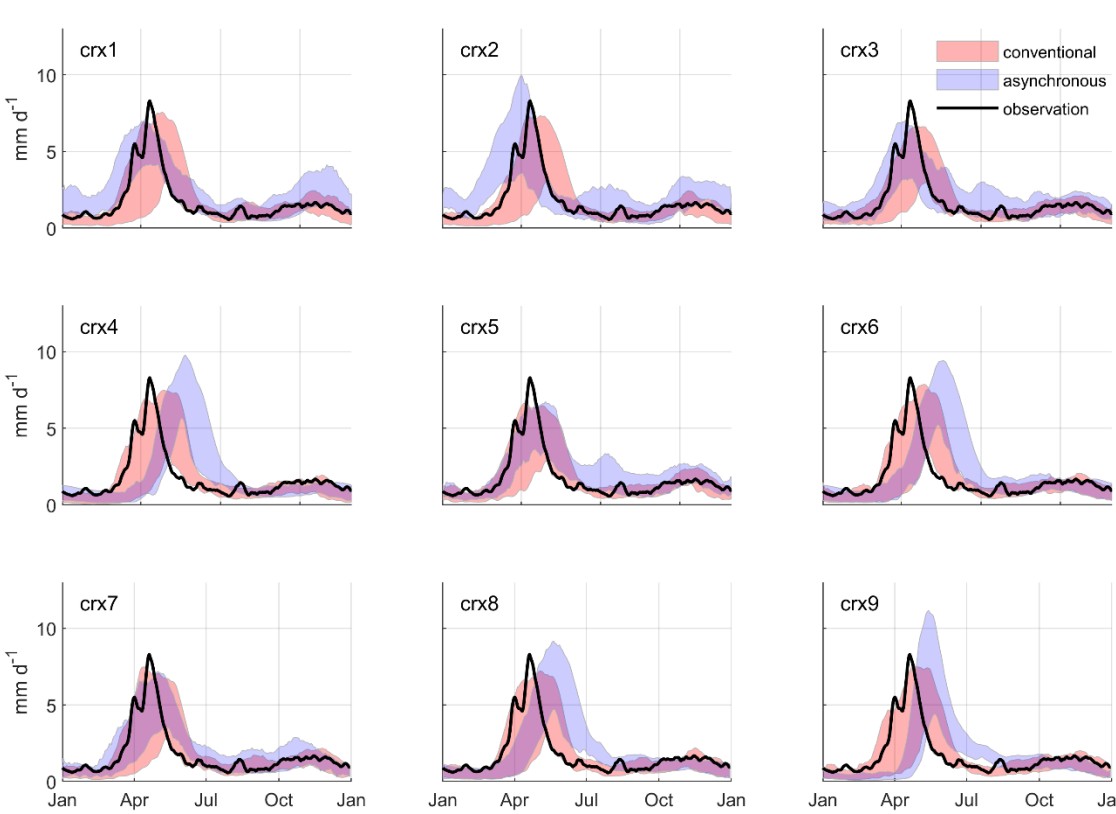


**Figure 6: Mean annual hydrographs simulated at site 2 over the reference period (1970 to 1999) for each NA-CORDEX simulation. Hydrographs produced by the conventional hydroclimatic modelling approach are compared to those produced by the proposed asynchronous workflow. Envelopes refer to the 10th and 90th percentiles out of the pool of 7 hydrologic models. A 5-day moving window is applied to enhance the signal to noise ratio. Corresponding observations are also illustrated.**





Figure 7 compares the hydrologic performance of NA-CORDEX simulations (crx1 to crx9) issued by the conventional modelling approach and the proposed asynchronous workflow at sites 1 to 4. Performance is sorted according to the RMSE value between simulated mean annual hydrographs and corresponding observations over the 1970-1999 reference period. The median RMSE value out of 7 hydrologic model simulations is here presented. Results first confirm the systematic capacity of the conventional modelling approach to provide a more accurate representation of the inter-annual hydrograph, corresponding

RMSE values ranging from ~0.9 to 1.2 mm d⁻¹. The performance issued by the conventional approach is also notably comparable from one site to another. On the other hand, the asynchronous workflow produces a systematic less accurate representation of the mean annual hydrograph. Most performing simulations (ranks 1 to 5) are affected by RMSE values ranging from ~ 1.3 to 1.6 mm d⁻¹, which can be considered as comparably performant relative to the conventional approach. A marked degradation is however observed for other less performing simulations (ranks 6 to 9, RMSE reaching ~2.5 to 3.0

mm d⁻¹ depending on the site). Sorting NA-CORDEX simulations according to their hydrologic performance systematically points out to same discrimination between the pool of most performing simulations (crx1, crx2, crx3, crx5 and crx7) and the less performing ones (crx4, crx6, crx8, and crx9).

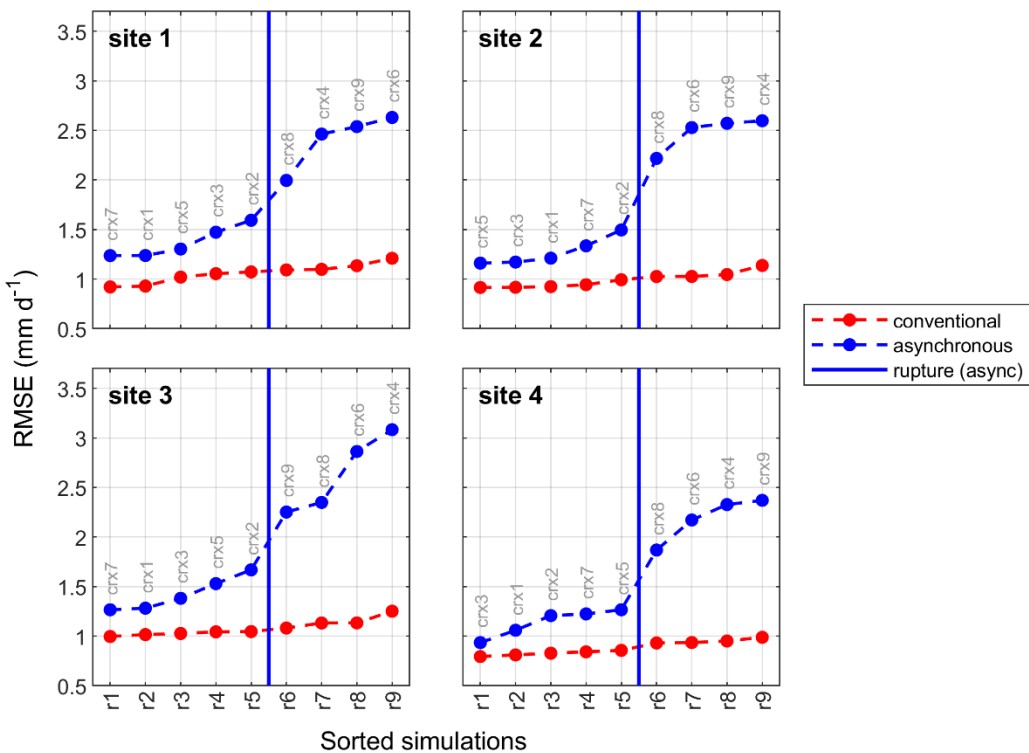

**Figure 7: Sorted hydrologic performances of NA-CORDEX simulations over the reference period at sites 1 to 4 for the conventional**
**hydroclimatic modelling approach and the asynchronous workflow. Performance is evaluated using the RMSE value between simulated mean annual hydrographs and corresponding observations. The median RMSE value out of 7 hydrologic model simulations is here presented. A rupture can be observed after ranks 5 for the asynchronous workflow.**





Figure 8 compares projected changes of the seasonal mean flows simulated by the conventional hydroclimatic modelling approach and the proposed asynchronous workflow at sites 1 to 4. Changes are expressed in relative terms (percentage) for the nival (DJFMAM) and the pluvial (JJASON) regimes and grouped according to the driving NA-CORDEX simulation (boxes). The top-5 most performing NA-CORDEX simulations identified for the asynchronous workflow are highlighted in blue. A group of outlying changes (all related to hydrologic model 1) projected by the conventional approach at site 3 is also identified in red. NA-CORDEX simulations being analysed separately, Figure 8 shows discrepancies in change values from one modelling approach to another. Site 2 being given as an example, the spread of changes projected by crx2 is noticeably reduced using the asynchronous workflow in comparison to the conventional approach. A shift in the projected direction of change for the pluvial mean flow ($\Delta_{\text{JJASON}}$) can also be observed, from a plausible decrease in the case of the conventional approach to a very likely increase for the asynchronous workflow. On the other hand, Figure 8 also shows that both approaches lead to comparable interpretation if the projected changes are analysed as an ensemble. Site 2 once again being given as example, both approaches strongly agree in projecting an increase of the nival mean flow ($\Delta_{\text{DJFMAM}}$). Both approaches also agree in projecting a decrease of the pluvial mean flow ($\Delta_{\text{JJASON}}$), except for a portion of projections mostly related to the crx2 simulation. The fact that the interpretation of the projected changes remains equivalent for both approaches can be generalized to all sites.

Figure 8 shows that the asynchronous workflow tends to provide more outlying changes values in comparison to the conventional approach. For all sites, numerous projections indicate very strong increases of the nival mean flow ($\Delta_{\text{DJFMAM}}$), reaching up to ~+100%. Such outlying projected changes are however systematically related to the less performing NA-CORDEX simulation identified at Figure 7. The sub-ensemble of change values resulting from the selection of most performing simulations (blue boxes) provides a reliable interpretation of the hydrologic changes with regard to the conventional approach, here considered as the benchmark. The conventional approach can also produce notable outlying change values of nival mean flow in the specific case of site 3, all related to hydrologic model 1. Changes of the seasonal high flows and low flows projected by both approaches are presented in Appendix B.





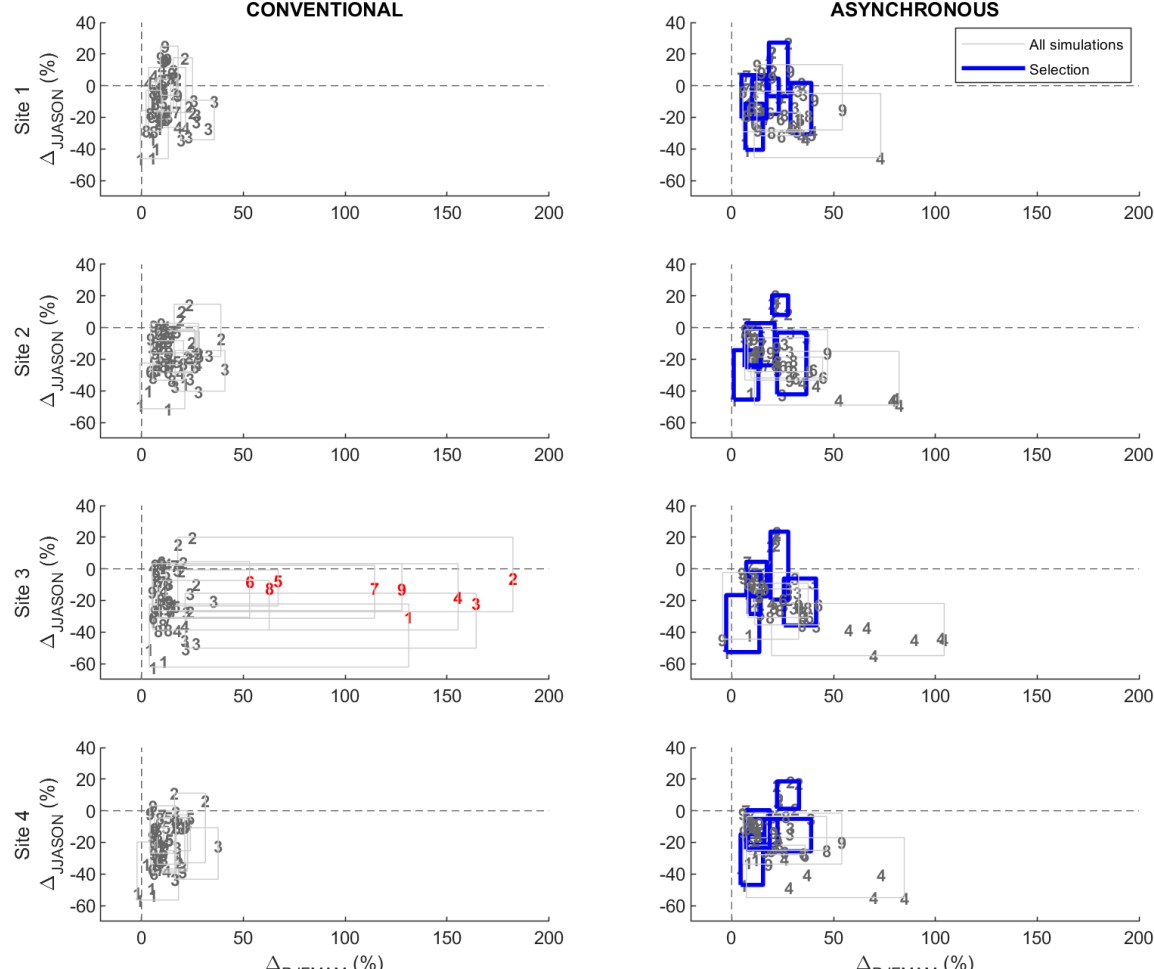

**Figure 8: Changes of seasonal mean flows ($\Delta_{\text{DJFMAM}}$ vs $\Delta_{\text{JJASON}}$) projected for sites 1 to 4 by the conventional hydroclimatic modelling approach and the proposed asynchronous workflow. Changes are expressed in terms of relative change (%) from 1970-1999 to 2040-2069. Change values are grouped according to the forcing NA-CORDEX climate simulation (boxes, crx1 to crx9). Blue boxes refer to the selection of most performing simulations produced by the asynchronous workflow. The red numbers refer to outlying changes projected by the conventional approach at site 3 with the hydrologic model 1.**

Table 4 summarizes the distributions of change values projected by the conventional modelling approach and the proposed asynchronous workflow. Results are displayed for site 2 using 6 hydrological indices describing seasonal (DJFMAM vs JJASON) mean, high, and low flow. High and low flow indices are computed based on annual maximal (and minimal) values considering a 2-year return period. Distributions are composed by all possible combinations between NA-CORDEX simulations and hydrologic models (n=63) for the conventional approach and by the selection of the top-5 most performing





simulations for the asynchronous workflow (n=35). Change distributions are described using the «direction» of change (the portion of values pointing out to an increase of a given index), the median value and its standard deviation.

Results first confirm a strong agreement between both approaches interpreting the changes of mean flow indices, the projected
increase being equivalent in terms of direction (98% vs 100%), median values (+15% vs +16%) and standard deviation (9% vs 8%). Both approaches also agree, but to a lesser extent, on the projected decrease of pluvial mean flow. The direction (10% vs 23%), the median change value (-17% vs -10%), and the standard deviation (14% vs 17%) lead to comparable interpretation of the change signal.

Modelling approaches do not agree as strongly in projecting high flows. While the asynchronous workflow indicates a probable
increase of nival high flows (direction=77% and median=+10%), the conventional approach rather provides a blurred signal. The direction of change (41%) indicates a weak consensus among projections and the median change value is small (+1%). In this case, standard deviation is comparable between both approaches (11% vs 12%). The opposite situation is observed for pluvial high flows where the conventional approach projects a probable decrease (direction = 21% and median = -13%) and the asynchronous, a distorted and vague change signal (direction = 40%, median = -2%, standard deviation = 23%).

Modelling approaches agree on the direction of change for low flows. They both indicate a probable increase of the nival lows flow (79% vs 77%) and a probable decrease of the pluvial low flow (6% vs 16%). The conventional approach however suggests a more severe increase of the nival low flow (median value = +56%) relative to the asynchronous workflow (+21%). The spread of the distribution is notably high in the case of the convention approach (+132%). Both approaches finally roughly agree on median change values (-18% and -12%) and standard deviations (18% and 12%) for pluvial low flows.

**Table 4. Interpretation of change value distributions projected by the conventional hydroclimatic modelling approach and the proposed asynchronous framework at site 2. The analysis is conducted on seasonal (DJFMAM vs JJASON) mean, high and low flows indices. High and low flows indices refer to the 2-year return period maximal (minimal) annual streamflow value.**

| INDICES | SEASON | DIRECTION OF CHANGE (%) | | MEDIAN CHANGE (%) | | STANDARD DEVIATION (%) | |
|---|---|---|---|---|---|---|---|
| | | CONV* | ASYNC** | CONV | ASYNC | CONV | ASYNC |
| Mean flow | DJFMAM | 98 | 100 | +15 | +16 | 9 | 8 |
| | JJASON | 10 | 23 | -17 | -10 | 14 | 17 |
| High flow | DJFMAM | 41 | 77 | +1 | +10 | 11 | 12 |
| | JJASON | 21 | 40 | -13 | -2 | 16 | 23 |
| Low flow | DJFMAM | 79 | 77 | +56 | +21 | 132 | 31 |
| | JJSAON | 6 | 16 | -18 | -12 | 18 | 12 |

*All NA-CORDEX simulations (n=63).*
** *Selected simulations based on hydrologic performance (n=35).*





### 4.3 Construction of hydrologic scenarios


Figure 9 illustrates change factors ($\phi$) computed as prescribed by Eq. (4) issued by the asynchronous modelling framework, displayed for each streamflow quantile at site 2. Change factors are computed on an annual basis (all data, no subsampling of the annual cycle) and for the nival (DJFMAM) and pluvial (JJASON) regimes that both experience low and high flow periods. Change factors are defined from percentile 0.5 to percentile 99.5 by increments of 1 (100 nodes), interpolated linearly. Resulted

are shown for all NA-CORDEX simulations and for the selected most performing ones, respectively. Annual factors depict little projected changes in streamflow quantiles from the reference to the future period. They confirm an increase for lower quantiles ($\phi$ roughly ranging between 0.9 and 1.5), while no clear change signal can be observed for higher quantiles. On the other hand, nival change factors (DJFMAM) show much more marked projected changes from reference to future. While all simulations agree on an increase for smaller streamflow quantiles ($\phi$ ranging between 1 and 2), $\phi$ reaches the value of 2.9 for

quantile 0.8. $\phi$ abruptly decrease for quantiles above 0.9, ranging between 0.9 and 1.3. Pluvial change factors (JJASON) are not as marked as nival factors. They confirm however a consensual decrease for quantiles below 0.8. The consensus weakens for higher quantiles, corresponding $\phi$ values being centered around 1 and affected by a larger spread. One must notice that the selection of NA-CORDEX simulations based on hydrologic performance typically agree with the ensemble composed by all simulations, except for projecting nival streamflow quantile from 0.5 to 0.9. In this case, the selected simulations provide a

much smaller increase of nival high flows, $\phi$ typically below 1.5.

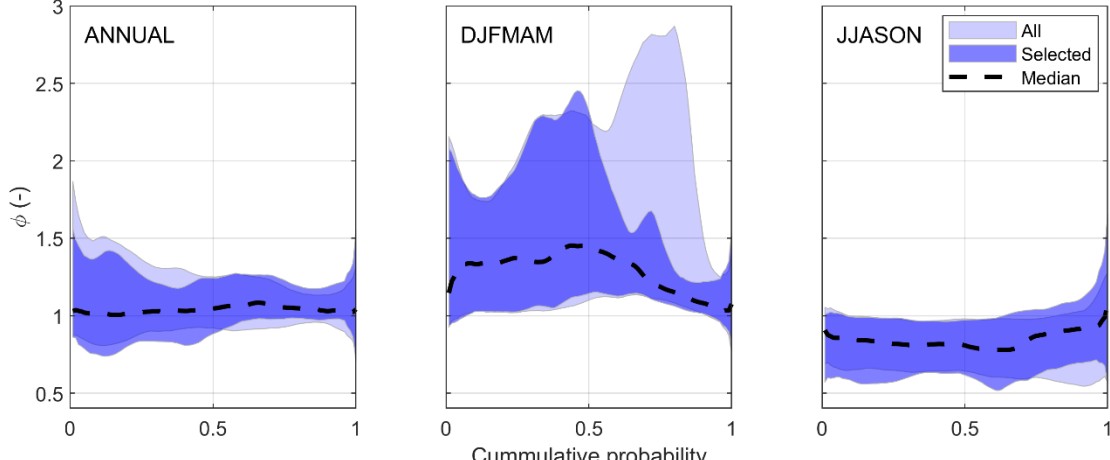

**Figure 9: Streamflow change factors ($\phi$) from the reference period (1970-1999) to future (2040-2069) issued by the asynchronous modelling workflow at site 2. Factors are computed on annual and seasonal (DJFMAM vs JJASON) bases. They are also presented for all NA-CORDEX simulations, and for the top-five selection of simulations based on hydrologic performance. Envelopes refer to**
**the 10th and the 90th percentiles. The median of the selected ensemble is also shown.**





Figure 10 displays the hydrologic scenarios produced over the Chaudière River subcatchments by applying the quantile perturbations to the observed streamflows for the selected years 1982, given as an example. The hydrologic scenarios reflect the relative changes embedded within the distributions of change factors shown in Figure 9. Future winter low flows are systematically higher relative to the observations. Mid-amplitude spring high flows are also affected by notable increases,
which is not systematically the case for high-amplitude peak flows. Summer low flows tend to decrease, while summer and autumn high flows are affected by moderate increases and decreases, depending on the climate scenario.

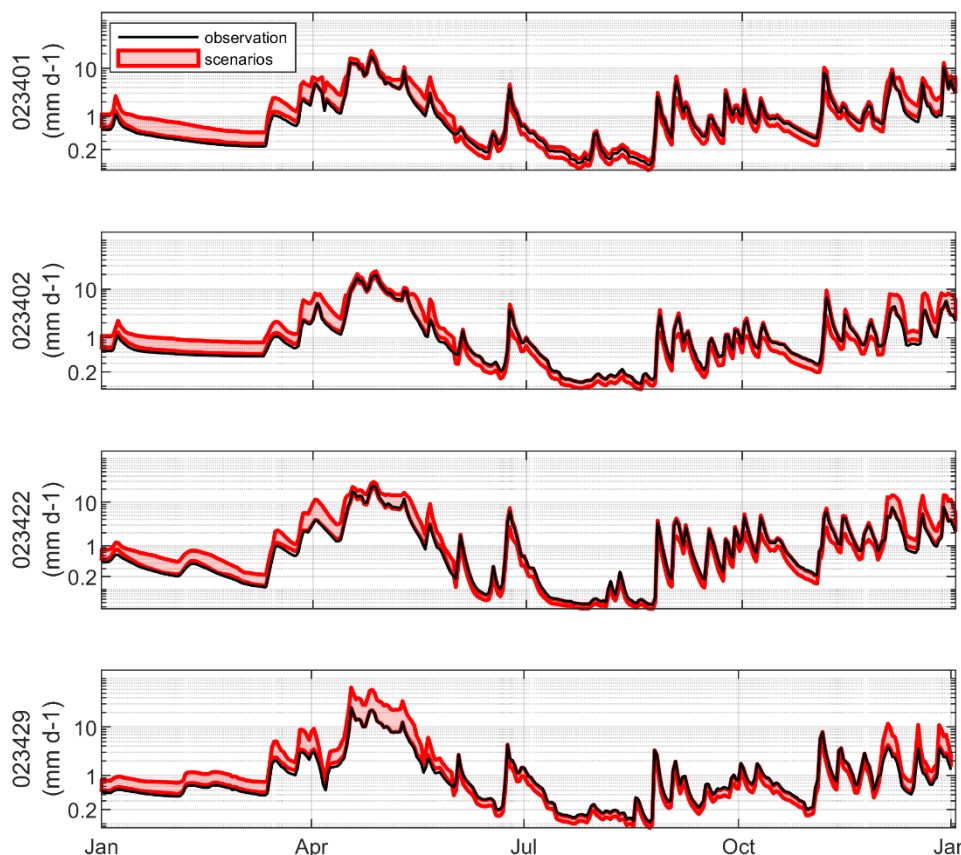

**Figure 10: Hydrologic scenarios (in red) produced by applying quantile perturbation to streamflow observations (in black) for each Chaudière River subcatchments (sites 1 to 4) for 1982 (given as example). The min-max red envelope refers to the nine scenarios**
**issued by the raw NA-CORDEX simulations. Note the log axis on the y axis.**





## 5 Discussion

### 5.1 A simplified, reliable, and advantageous hydroclimatic modelling workflow

Nowadays, the quantification of climate change impacts on water resources mostly resorts to the implementation of top-down
modelling cascades, translating climate model outputs into simulated hydrologic time series at the catchment scale. Typically,
a statistical post-processing is applied to the raw climate model outputs in order to reduce biases imbedded in the simulated
climate variables. Hydrologic models are also typically calibrated when forced by meteorological observations aiming to
identify optimal parameter sets minimizing errors between simulated and observed discharge at a given catchment outlet.
Assessing the impact of climate change on the hydrologic regime of a catchment using this conventional modelling approach
presents numerous drawbacks documented in the scientific literature: (1) the statistical post-processing of climate model
outputs may disrupt the physical consistency between the simulated climate variables and even alter the corresponding trends
from a reference period to a future period; (2) the modelling work flow is relying highly on the availability and quality of
meteorological observations in order to conduct the statistical post-processing of climate model outputs and the calibration of
the hydrologic model; and (3) it also requires a high level of expertise and computing capacities to postprocess the outputs and
using the complex statistical methods, restraining the participation of end-users in interpreting and attributing confidence to
the simulation results.

In this study, we propose an innovative and quite straightforward asynchronous modelling workflow that enable the production
of hydrologic scenarios without resorting to the statistical post-processing of climate model outputs. This unconventional
approach is conducted calibrating the hydrologic model forced directly with raw climate model outputs instead of
meteorological observations, using an objective-function that exclude the temporal correlation between the observed and
simulated hydrologic responses. Calibrated hydrologic models allow the conversion of raw climate model into corresponding
reference and future simulated hydrologic responses. Hydrologic scenarios are subsequently produced by applying quantile
perturbations to available streamflow observations, change factors being defined by relating simulated reference and future
hydrologic responses for each streamflow quantiles. Quantile perturbation is applied to simulated climate variables such as
precipitation or reference evapotranspiration (Ntegeka et al., 2014), but never before, to our knowledge, to the simulated
hydrographs resulting from a hydroclimatic modelling cascade.

We validated the proposed asynchronous workflow by comparing its resulting projections of the hydrologic regime with a
conventional hydroclimatic modelling approach. As shown by others (e.g. Muerth et al. 2013), our results confirmed the
capacity of post-processing of raw climate model outputs to increase the performance of the hydrologic response simulated
over the historical reference period. On the other hand, our results demonstrated that the projected changes of the seasonal
mean flows, taken as ensembles, converged to equivalent conclusions disregarding the modelling approach. The concordance
between both approaches did not occur as sharply for high and low flow indices, suggesting further investigations would be





required to clarify how and to which extent the projection of extreme values is sensitive to the selection of the hydroclimatic modelling approach. We here emphasize the fact that the asynchronous workflow is vulnerable to strong biases affecting raw climate model outputs and is consequently more prone in producing outlying projections of hydrologic indices. However, the performance of the simulated response over the reference period provided a functional criterion to identify less performing NA-CORDEX simulations. Based on the results shown in Section 4, we would advocate for the exclusion of these simulations for the analyse of the simulated projections of the hydrologic regime using an asynchronous modelling framework.

As opposed to conventional hydroclimatic modelling, the proposed workflow presents numerous benefits: (1) it increases confidence in the hydrologic scenarios since it is conducted with raw climate model outputs, thus preserving physical consistency between simulated climate variables and original trends simulated by the climate models – some authors also argue that raw climate model outputs are expected to improve in resolution and reliability with time (e.g. Teng et al., 2015, Chen et al., 2017); (2) it does not resort to meteorological observations, nor for operating statistical post-processing, nor for calibrating the hydrologic model, facilitating assessment of climate change impact on water resources for regions afflicted with observation scarcity – we would also argue that our approach does not inject uncertainty into the modelling cascade from the intrinsic limitations of post-processing methods (Laux et al., 2021), nor from poor quality observations or reference product describing the reference climate system (Hwang et al., 2014; Kotlarski et al., 2017); (3) it is simple to implement and lighter in computing requirements.

## 5.2 A bottom-up perspective

Statistical post-processing of climate model outputs implies a necessary trade-off between key methodological benefits and drawbacks in the scope of providing reliable and supportive information for adaptation to climate change. On one hand, simulated climate variables are corrected to fit statistical properties of the observed climate system. On the other hand, statistical post-processing disrupts physical consistency and alters trends in the simulated climate variables. While designing statistical post-processing, a decision is implicitly taken on how these benefits and drawbacks are weighted. In a pure top-down perspective, statistical post-processing is applied according to climate-oriented prerogatives, the end-user rarely being involved in deciding upon which benefit to be prioritized and which drawbacks to be limited. Moreover, not communicating source biases affecting raw climate model outputs constrains the capacity of impact modelers and end-users in assessing the climate model representativeness and attributing confidence to resulting climate scenarios. Nowadays, solutions explored by the scientific community mostly resort to the development of sophisticated post-processing methods. Even though such approaches present undeniable benefits in terms of post-processed physical consistency and trend preservation, we would argue that they further enlarge the gap between climate specialists and water resources end-users, and somewhat restrain adaptation to climate change.





The approach proposed in this study remains in essence a top-down modelling workflow. Through notable simplifications, straightforward constructions between raw climate model outputs and impact models, this alternative framework creates a
space for an increased participation of impact modelers and end-users in interpreting climate change impacts on water resources (Ehret et al., 2012). It is thus compatible with integrated and transdisciplinary environmental assessments and modelling frameworks in support to decision and policy making (Hamilton et al., 2015; Rössler et al., 2019). By translating raw climate model outputs into corresponding simulated hydrologic responses, the representativeness of climate models can be assessed in a language impact modelers and end-users can better understand. Based on the simulated hydrologic responses over the
reference period (see Mudbhatkal and Mahesha, 2018), key methodological questions can be addressed and debated through an open and empowered dialogue with climate specialists. Questions such as: Are climate outputs representative enough to assess the impacts of climate change on water resources? Should the climatic or hydrologic representation be prioritized? Or both? How should less representative simulations be treated? Rejected, weighted (e.g. Shin et al., 2020), or considered equal. Are scenarios required for the adaptation to climate change, or relative change signals sufficient? Should post-processing be
applied to raw climate model outputs? We believe decisions upon such questions require a sound understanding of simulated climate forcing, but also an in-depth awareness to requirements and local specificities of the hydrologic system exposed to climate change.

### 5.3 Limitations

We believe the experimental design shown in this study is sufficient to set a proof-of-concept and demonstrate the applicability
of the proposed workflow, at least for seasonal mean flow indices. Verification confirmed that the simulated hydrologic response issued by the proposed asynchronous workflow is affected by systematic errors (or hydrologic biases), mostly notable in terms of synchronism of the mean annual hydrograph during spring flood. Considering this, we would advocate that the proposed workflow should be used with caution if focusing on analysing extreme events. To formally assess the impact of climate change on a given domain, however, a larger ensemble of climate simulations should be considered. Since the
workflow does not involve statistical processing of climate model outputs, we would recommend the use of high resolution over coarse gridded climate simulations in order to rely on an improved representation of local scale processes. The use of seven conceptual lumped hydrologic models can also considered as a limitation to our approach. Although they provide a diversity in modelling structure, no formal evaluation of this specific source of uncertainty as been considered in this study (calibration metric, calibration period, structure complexity).

Assessing the impact of climate change on water resources within the proposed framework implies that the resulting hydrologic scenarios are inevitably tainted with (hydrologic) biases. These biases emerge form raw climate model outputs, but also from the limitations imposed by the structure of the hydrologic models. We believe further work should focus on evaluating how theses two sources are intertwined and how parametric compensation affects the trade-off between hydrologic scenarios fitted to observations and the preservation of the hydrologic change signal embedded within raw climate model outputs. In the





meantime, we would argue that parametric compensation should be minimized as much as possible to preserve the hydrologic change signal. This could be achieved, for example, by restraining parametric spaces during calibration as close as possible to realistic boundaries or favoring physically based descriptions of hydrologic processes. Even if climate models constantly improve, we raise attention that corresponding biases can still be important – a judgment must be made in order to attribute confidence to resulting hydrologic scenarios. Chen et al. (2021) explicitly raised the idea of an optimal selection of climate

simulations before producing hydrologic scenarios to cope with their limitations in representing local hydrometeorological patterns. We do not propose here any specific guidelines except that such attribution must consider the scope and objectives of the conducted study and should involve as much as possible climate specialists, impact modelers and end-users.

The proposed workflow is not limited by available meteorological observations, but to available streamflow observations. To assess the impact of climate on ungauged water resources, modelers can translate the hydrologic perturbation signals under

the assumption of representativity of available discharge observation with regards to the ungauged domain. If ungauged streamflow is estimated before applying change factor (using area ratio, hydrological modelling, or optimal interpolation), corresponding uncertainties must by considered.

Constructing hydrologic scenarios using quantile perturbations, our results demonstrated the necessity in identifying a suitable time period to define change factors. Such resolution must consider specificities of the local flow regime magnitudes. The

identification of an optimal duration remains an open question, keeping in mind the use of a moving window could become necessary to compensate breakpoint in the hydrologic scenarios. Even considering the relative change for each streamflow quantiles, the capacity of quantile perturbation to preserve mean flows and seasonal budgets should further be explored and assessed. Further work should also focus on formally comparing our approach to a conventional hydroclimatic modelling framework involving statistical post-processing, analysis the preservation, and consistency for simulated hydrologic variables.

## 6 Conclusion

This study explores an innovative and straightforward hydroclimatic modeling workflow enabling the construction of hydrologic scenarios without meteorological observations. Hydrologic models are forced with raw climate model outputs and calibrated using nCPRS, an objective function that exclude the day-to-day temporal correlation between simulated and observed hydrographs. Hydrologic scenarios are produced applying quantile perturbation to the available observed streamflow

measurements. This workflow is implemented over a mid-scale catchment located in southern Québec, Canada using an ensemble of NA-CORDEX simulations and a pool of lumped conceptual hydrologic models. The asynchronous workflow is validated by comparing its resulting projections of hydrologic indices with a conventional hydroclimatic modelling approach. The latter involved post-processing of raw climate model outputs and calibration of hydrologic model using meteorological observations. Results showed that both methods lead to equivalent projections of the seasonal mean flow indices. Both

approaches did not agree as well projecting high and low flow indices, suggesting further works should be conducted to confirm



the reliability of the proposed workflow to assess the impact of climate change on extreme hydrologic events. Results also highlight the importance of considering seasonal fluctuations of the hydrologic regime while applying quantile perturbations to the observed streamflow measurements. We argue that the suggested workflow increases the confidence attributed to the hydrologic scenarios, mostly because it preserves physical consistency between driving simulated climate variables. We also

underline that the workflow ease communication between climate experts, impact modelers and end-users, thus supporting decision making in the process of the adaptation of water usages to climate change.





## Appendix A

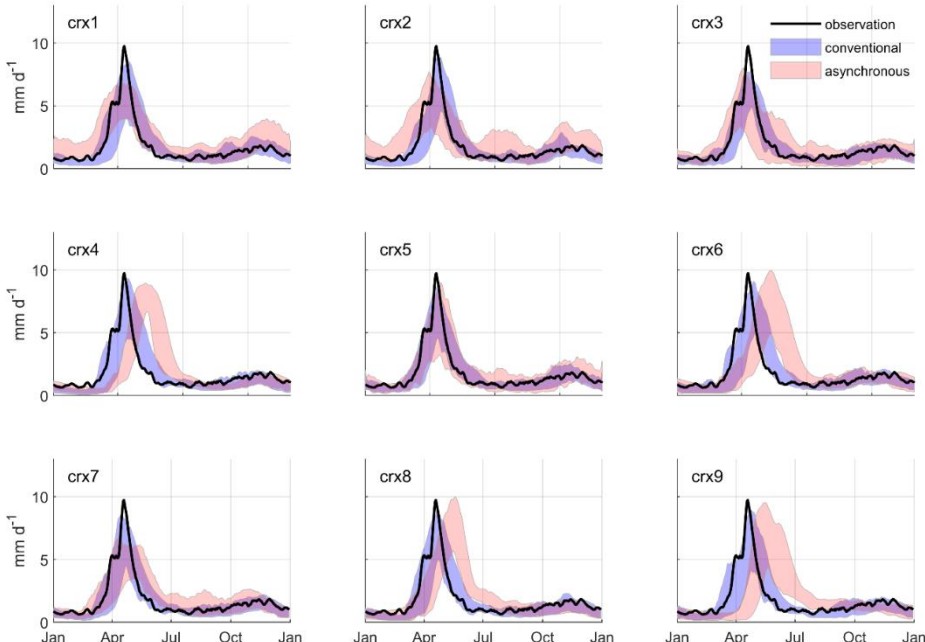

**Figure A1: Mean annual hydrographs simulated at site 1 over the reference period (1970 to 1999) for each NA-CORDEX simulation. Hydrographs produced by the conventional hydroclimatic modelling approach are compared to those produced by the proposed asynchronous workflow. Envelopes refer to the 10th and 90th percentiles out of the pool of 7 hydrologic models. A 5-day moving window is applied to enhance the signal to noise ratio. Corresponding observations are also illustrated.**





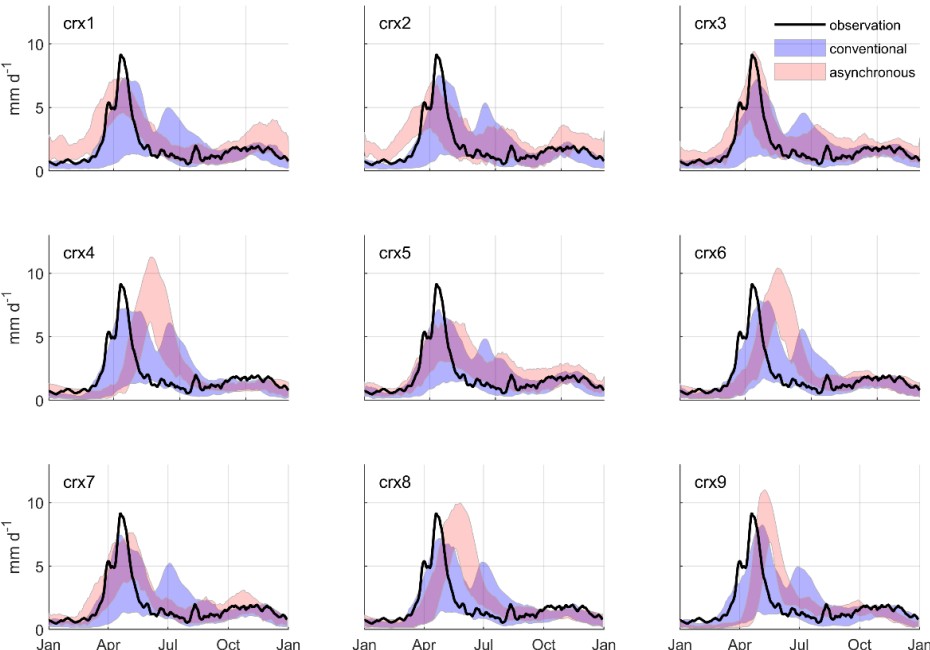

**Figure A2: As Figure A1 but for site 3.**

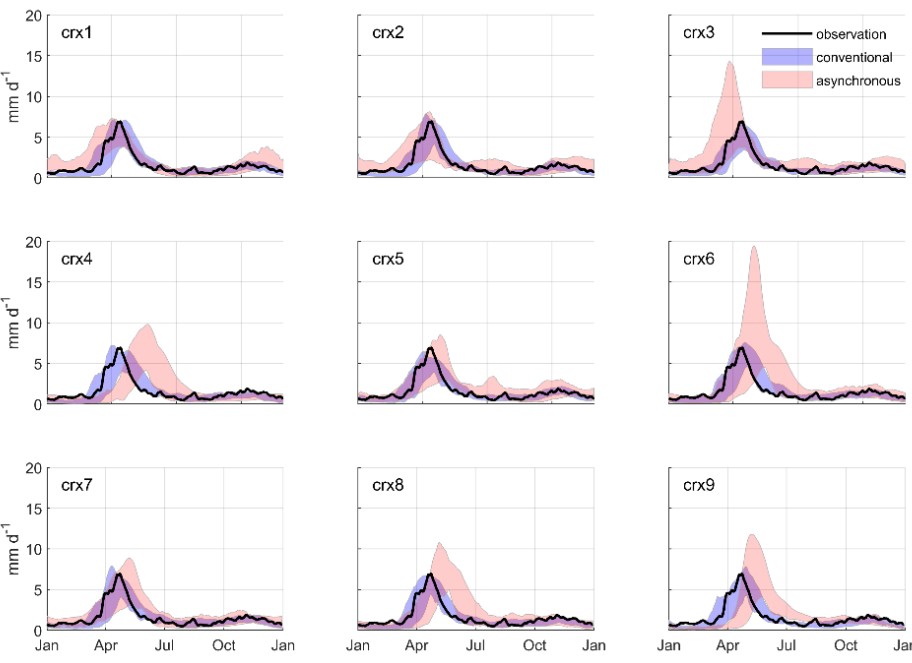

**Figure A3: As Figure A1 but for site 4.**





## Appendix B

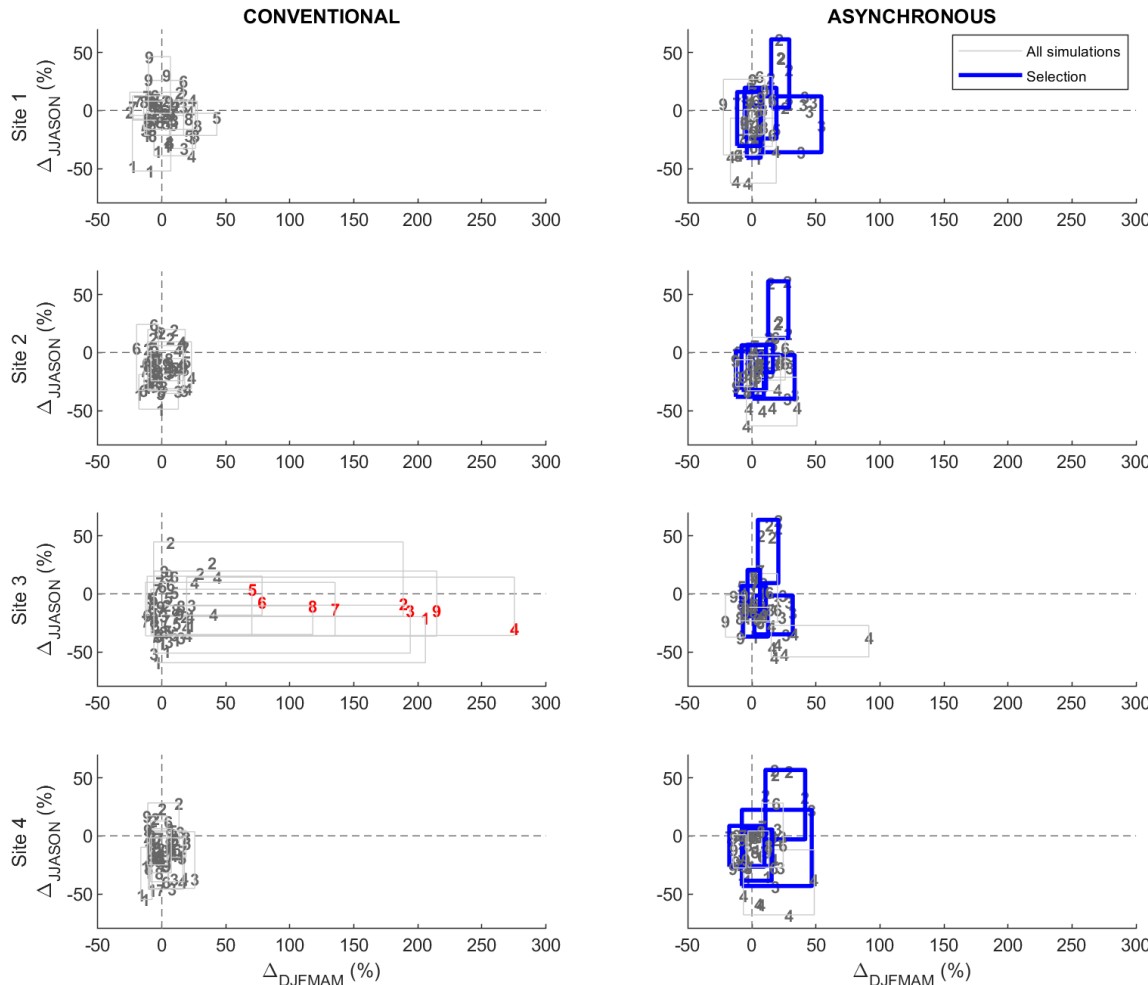

**Figure B1: Changes of seasonal high flows ($\Delta_{DJFMAM}$ vs $\Delta_{JJASON}$) projected for sites 1 to 4 by the conventional hydroclimatic modelling approach and the proposed asynchronous workflow. High flows are computed based on the annual maximal value with a 2-year return period. Changes are expressed in terms of relative change (%) from 1970-1999 to 2040-2069. Change values are grouped according to the forcing NA-CORDEX climate simulation (boxes, crx1 to crx9). Blue boxes refer to the selection of most performing simulations produced by the asynchronous workflow. The red numbers refers to outlying changes projected by the conventional approach at site 3 with the hydrologic model 1.**





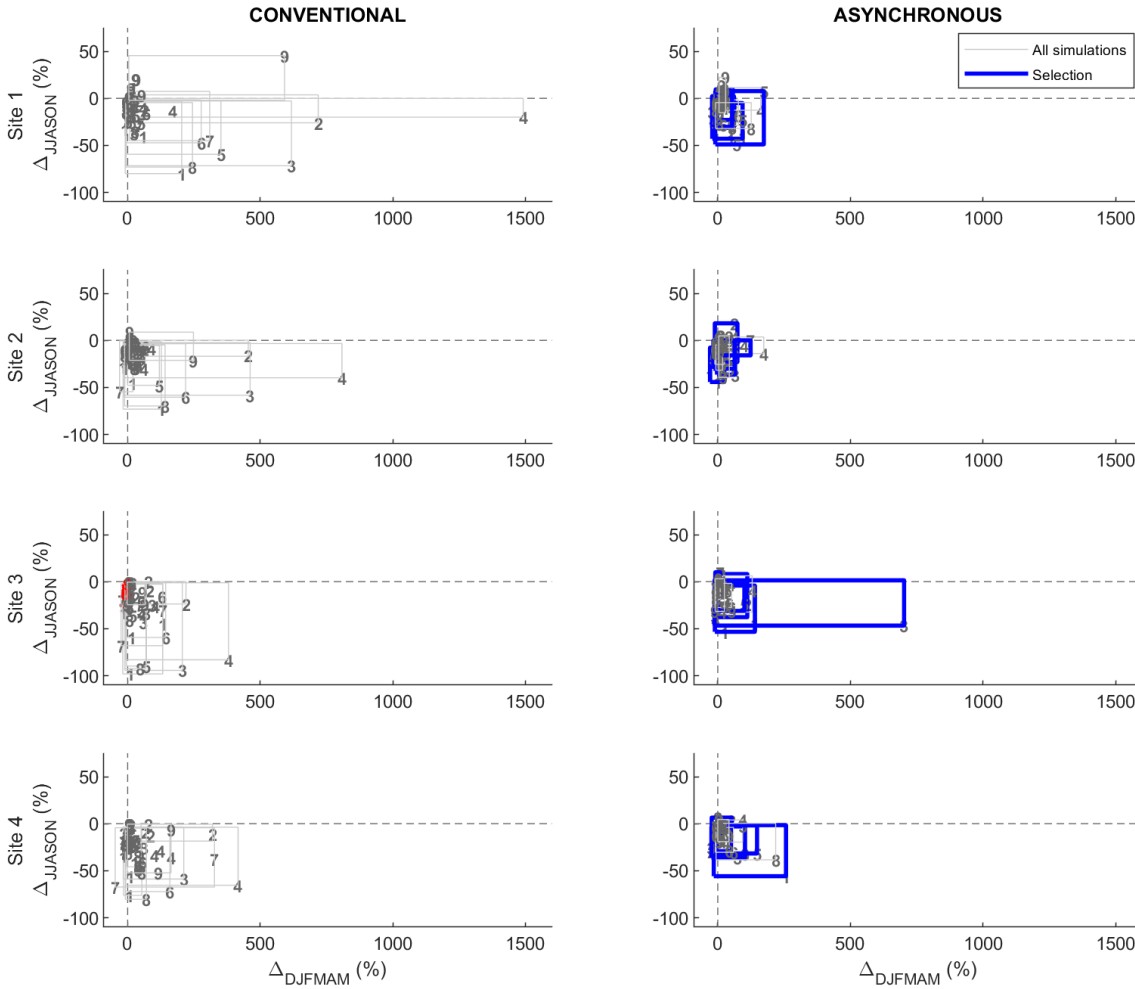

**Figure B2: As Figure B1 but for low flows.**




## Author contribution

SR designed the experiments and SR carried them out. SR and AT developed the model codes and SR performed the simulations. SR prepared the manuscript with significant contributions from co-authors.

## Competing interests

The authors declare that they have no conflict of interest.

## Acknowledgments

This research was funded by the Mitacs Accelerate program for scholarship to SR (IT12297) and by the French National Research Agency under the future investment program ANR-18-MPGA-0005. Authors acknowledge the financial contribution
of the Québec regional county municipalities of Beauce-Sartigan, Nouvelle-Beauce, and Robert-Cliche. We also acknowledge the World Climate Research Programme's Working Group on Regional Climate, and the Working Group on Coupled Modelling, former coordinating body of CORDEX and responsible panel for CMIP5. We also thank the climate modelling groups (listed in Table 1 of this paper) for producing and making available their model output. We also acknowledge the Earth System Grid Federation infrastructure an international effort led by the U.S. Department of Energy's Program for Climate
Model Diagnosis and Intercomparison, the European Network for Earth System Modelling and other partners in the Global Organisation for Earth System Science Portals (GO-ESSP). We also acknowledge Quebec Ministry of Environment and Fight Against Climate Change (MELCC) for meteorological and discharge data.

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
