# Peer review of "Producing reliable hydrologic scenarios from raw climate model outputs without resorting to meteorological observations"

_Hydrology and Earth System Sciences, 2022_

## Author Response (AR1)

**RC1**: 'Comment on hess-2022-264', Anonymous Referee #1, 29 Sep 2022
The paper entitled "Producing reliable hydrologic scenarios from raw climate model outputs without resorting to meteorological observations" by Simon Richard and co-authors proposes a new modeling workflow to derive hydrologic scenarios from climate model projections without resort to the usual step of bias correction of climate projections. The topic is well suited for the journal Hydrology and Earth System Sciences and I think the paper raises important and interesting questions at the interface of climate sciences and hydrology.

I found the paper well written and organized, the methods are described in details, and a case study allows readers to have an idea of the performance of the proposed framework for a typical application (in the present case: producing hydrologic scenarios under climate change for a catchment located in Québec - Canada). I enjoyed the comparison with a more conventional approach (i.e. including a step of bias correction of model outputs), and the way the results are displayed and discussed in Section 4: Results.

My only sticking point is the discussion (Sect. 5). In contrast with Section 4, I have the impression that the authors are overselling their approach in the discussion section. Indeed, the general tone of this section tends to make the reader forget that the conventional approach still outperforms the one proposed in the present paper (cf Fig. 6 and 7). I acknowledge the interest of the alternative approach proposed by the authors for both (1) complement the standard approach in areas where meteorological observations are available, and (2) allow hydrologists to perform hydrologic projections where meteorological observations are too sparse to enable reliable climate model bias correction. But I also think that the discussion is too harsh towards climate model bias correction (i.e. the conventional approach), and that the proposed approach does not solve many of the (legitimate) questions raised by the authors in the discussion. More particularly:

**We edited Section 5, reflecting the strengths and weaknesses of both approaches in a neutral way. Section 5.1 explicitly presents the proposed approach as a complement to conventional hydroclimatic modeling.**

* Integrating meteorological observations in the modeling chain. When meteorological observations are available, I think it is a shame to disregard them. Maybe the conventional approach of bias correction of climate outputs is not ideal (and the authors are right to talk about its limitations), but removing it without replacement is equivalent to do without all the information embedded in meteorological observations. Of course having a method to deal with poorly

gauged areas (in terms of meteorological variables) is a plus, but when data are available I don't see why not using them. So I think that the authors should acknowledge more clearly in the discussion that when meteorological data are sufficient for climate model bias correction, this method is still the one that performs best.

**Section 5.1 (lines 449-454) encourages a sound use of reliable meteorological observations when available. It also encourages further exploration of combined approaches (conventional-asynchronous) aiming to maximise available observations at the regional scale or for modelling hydrological processes using more complex physical descriptions.**

* Physical consistency of the modeling chain. I agree that bias correction methods disrupt the physical consistency of climate projections. But clearly the approach proposed in this paper does not provide any solution to this problem. The calibration of a hydrological model directly from raw climate outputs will mix climate biases and hydrologic biases (as acknowledged by the authors in Sect 5.3), which results in a completely non-physical hydrological model. So I think that the authors should acknowledge that both approaches are equally breaking the physical consistency of the hydro-meteorological processes involved in the models used to investigate the hydrological response of our environment to climate change, and that where we decide to do it (and unfortunately often to hide it) within the modeling chain is somehow a matter of taste.

**The perturbation of the physical consistency between simulated hydrologic processes through parametric compensation is explicitly acknowledged as a limitation in Section 5.3 (lines 501-505).**

* Interpretability of climate projections by end-users. The questions raised at the end of Sect 5.2 (l 465 - 472) are interesting and legitimate, but with a very few exceptions can also be addressed with bias-corrected climate model outputs. In a slightly different note, I do not think that bringing expertise in analyzing, selecting and pre-processing climate model outputs is in itself a bad thing. An intense and constructive discussion between climate modelers, statisticians performing bias corrections, hydrologists and stakeholders (I probably forget important participants) is of course essential, but I am a bit skeptical about the idea of a more direct (and therefore possibly less careful) use of raw climate model outputs. From my personal experience, the support from experts in climate models biases and bias correction is essential to avoid misuse of climate projections.

**We did not intentionally suggest excluding climate model experts in the analysis of bias. We rephrased line 473, which could be misleading regarding this aspect**.

To sum up, I think that the present manuscript is interesting and well written, and I my opinion deserves publication after revisions. My main concern is about the discussion, which I believe can be improved by moderate revisions, mostly by rephrasing this section in a more objective way.

Hereafter are some minor comments/questions I had during my reading of the manuscript:

- L119: how many RCM grid cells cover your study area?

**The information is added to the revised manuscript (line 120).**

- Fig 1: Maybe add information about topography.

**Figure 1 has been edited with topography.**

- Fig 2: It would be nice to compare with the conventional approach in the figure (i.e. have 2 workflows in the figure) not only in the caption.

**We believe that comparing both workflows would notably impact the conciseness of the manuscript. A detailed description of the conventional modelling approach is provided by Ricard et al. 2020. A reference has been added to caption of Figure 2 (line 159).**

- Throughout Sect. 4: also mention relative biases. For instance L 243: "Biases typically range between -1 and +2 mm/day (xx %) depending on ...".

**Relative biases computed on daily values are affected with a high variability, especially for precipitations. We converted the constant +0.5 mm/day bias (observed from January to August and from mid-October to December) into relative terms (+27%, see line 243) using the median of daily values over the corresponding period. We believe however that the min-max range (from -17% to +168%) might mislead the reader. We remind that a 5-day moving window is applied to all time series in Figures 3 and 4.**

- L 264 and after: Validation of the asynchronous -> I would prefer "assessment" instead of "validation" (maybe personal taste).

**As suggested, "assessment" was replaced "validation" in the revised manuscript (line 266).**

- L 298: "which can be considered as comparably performant relative to the conventional approach" -> one of the few places in sect 4 where you are not very fair with your results. Consider rephrasing.

**The sentence has been rephrased to: " [...] which is comparably performant relative to the conventional approach." (line 300).**

- L 369: typo in percentile definition: 0.5 -> 0.05

**The typo in percentile definition has been changed to 0.005 and 0.995, respectively (line 371).**

- L 386: maybe remind for which period and RCP scenarios the hydrologic scenarios are made.

**The description has been rephrased (lines 388-390), indicating the corresponding period and the RCP.**

- L422: we validated -> we assessed the performance

**" validated " has been changed to " assessed " (lines 426 and 535).**

- L 519-521: the way this paragraph is written gives the impression that you consider low and high flows as extreme events. Maybe consider rephrasing?

**"extreme events" has been changed to "high and low flow events".**

**Citation**: https://doi.org/10.5194/hess-2022-264-RC1

'Comment on hess-2022-264', Anonymous Referee #2, 15 Nov 2022

Review for "Producing reliable hydrologic scenarios from raw climate model outputs without resorting to meteorological observations". This manuscript seeks to provide a new framework that can use regional climate model projections (CORDEX) to provide reliable hydrological projections. This framework aims to avoid using meteorological forcing data. Although it is an important topic, I feel most of the claimed goals are not well supported.

**We acknowledge that the presentation of a novel methodological framework can raise doubts and suspicions. Our intention with this paper is to provide as much arguments as possible supporting the idea that the framework could be useful to hydrologist assessing the impact of climate change on water resource in a situation where meteorological observations are rare. We are fully aware that the scope of the paper only provides a proof of concept and a partial validation. We are confident however that further work could be conducted to provide a more in-depth comparison with conventional hydroclimatic modelling.**

1. Although the meteorological data is not used, it still requires streamflow observations. I agree that it still uses less data than "conventional" approaches. However, regions with poor meteorological data are less likely to have reliable streamflow observations as well. Therefore, the benefit of this approach is questionable.

**We do not fully agree with comment raised by RC2. An example, in Northern Québec, streamflow data are available at the outlet of some large catchments while almost no meteorological stations are located on the watershed, providing large uncertainty related to precipitation and 2m temperature. We believe, in such cases, that the implementation of an asynchronous modelling framework could provides notable benefits in comparison to conventional modelling.**

2. Following my comment above, I think the missing part is: under what meteorological forcing data uncertainty levels the proposed framework is more advantageous? For instance, if we have only one precipitation but good streamflow gauges (not sure if that is realistic), the proposed framework outperforms the conventional approaches.

**An assessment scheme comparing the performance of both frameworks using intentionally scarcer meteorological observations is suggested as further work in Section 5.1.**

The description of the method requires more details. For instance, in line 170: which parameters are calibrated to minimize nCRPS?

**As stated in section 3.3, the calibration is performed using the same objective function, calibration period, and configuration of the optimization algorithm. We specified in the revised version of the manuscript that the same model parameters are calibrated in both frameworks (line 224). However, we think that presenting every parameter for each model and their role would make this section wordy, without bringing key information to the study. A reference has been added to Table 3 for additional information on the model parameters.**

3. When we are using regional climate model projections/simulations, we tend to be more interested in the long-term statistics, e.g., trends, standard deviations etc. However, only long-term climatology is discussed.

**We limited the analysis to long term climatology for conciseness of the paper.**

4. In the title, "scenarios" and "climate models" make me automatically think about climate change and long-term trends. However, these perspectives are not discussed and/or validated. I would suggest the authors to modify the title and the manuscript to avoid any confusions.

**We believe that the title fits the scope of the paper. We would like if the reviewer could highlight better the confusing elements. Since future climate and hydrologic conditions cannot be directly validated, a diversity of sound methods validated over the reference historic conditions appears to us as the best compromise to attribute confidence in hydrologic scenarios.**

5. Finally, I think it should present the optimized parameters to see if they are physically reasonable.

**We did not include a detailed description of the parameter values because the physical interpretation of the global conceptual hydrological model parameters is difficult. Only few parameters could be related to measurable physical quantities. Additionally, the calibration algorithm (the Shuffle Complex Evolution) requires the specification of an upper and lower bound. The parameter values will necessarily lie within the bounds that are forced. Therefore, parameters cannot get a value that is of an order of magnitude different from the ones obtained with a "regular calibration".**

**Finally, one should expect the parameter values obtained from an objective function that does not consider the temporal correlation to differ from the ones found with a more traditional score like RMSE or KGE. This makes the comparison of the two sets of parameters complex as the same hydrologic model is expected to behave in a different way in an asynchronous fashion.**

**Citation**: https://doi.org/10.5194/hess-2022-264-RC2

---

## Author Response (AR2)

**RC**: ['Comment on hess-2022-264'](), Anonymous Referee #3, 27 Feb 2023
Projection of the streamflow changes is vital for the climate adaptation and mitigation. Simon Richard and co-authors propose a new modeling workflow to provide streamflow projections without resort to the usual step of bias correction of climate projections. I went through the precious two anonymous referees' comments and read the manuscript carefully. In general, I agree with the reviewer #2. The topic is important, but the current manuscript is not strong enough. My comments are below:

1.The author pointed out numerous weaknesses of the traditional approach including the " the statistical post-processing of climate model outputs may disrupt the physical consistency between the simulated climate variables and even alter the corresponding trends", " the modelling work flow is relying highly on the availability and quality of meteorological observations" and " it also requires a high level of expertise and computing capacities to postprocess the outputs and uses non-trivial statistical methods".

However, whether these weaknesses can be solved by the new approach is doubtful. For example, the new approach still needs model calibration, hydrological model simulation and projection, and the construction of hydrologic scenarios using quantile mapping and correction. It still requires a high level of expertise and computing capacities to postprocess.

**We get the point. We all agree on the drawbacks and limitations affecting traditional hydroclimatic modelling approaches involving bias correction/post-processing of raw climate model outputs, namely the disruption of physical consistency, a strong requirement for observations and a high (still increasing) level of complexity. The latter are well documented in the scientific literature. We also fully agree with Reviewer #3 that the proposed asynchronous framework does not completely solve all the weaknesses of the traditional approach. We rather introduced the framework as a complementary analysis tool that can provide a sound alternative to impact modelers considering that (1) it preserves the physical consistency of climate model outputs (although it still requires the calibration of a hydrologic model, discussed below), (2) it requires no meteorological observations (a significant benefit since most of the earth system is affected by data scarcity), and (3) it involves less modelling processes (post-processing is exclusively applied to streamflow instead of numerous climate variables such as precipitation, air temperature, but also air humidity, radiations and wind speed, in some cases). We also discussed how impact modellers should ponder the use of the proposed framework weighting**

**corresponding drawbacks and benefits, within the scope and aims of a given study. Ultimately, we believe that while our approach requires further test and validation, it is innovative, relevant to the scientific community and worthy of publication. We remain fully available to discuss these issues with Reviewer #3. Comments and notifications have been added to the manuscript [lines 453-464] to summarize the above arguments.**

Moreover, the author complain that traditional approach needs bias-correction before hydrological simulation, while the new method does not. I am confused that, why we cannot directly calibrate the hydrological model using the raw climate output from GCM/RCM, and then use the future meteorological forcings to perform hydrological modeling? By doing this, the systematic errors in meteorological forcings are also bias-corrected (during the hydrological model calibration) and no additional meteorological observations are needed.

**We indeed calibrate the hydrologic model using the raw climate output from RCM over the reference period. We also subsequently use the "future" raw RCM forcings to perform hydrological modeling (using a model calibrated over the reference period), thus projecting the future hydrologic response corresponding to raw RCM forcings. At anytime, however, we correct/post-process systematic errors in RCM forcings. We could then consider that the calibration of the hydrologic model corrects systematic errors of the resulting simulated hydrologic response, but not the raw RCM forcings, although both are obviously related. We hope these clarifications are valuables to Reviewer #3.**

In addition, the third comment of Reviewer #2 is not well answered, leading to a doubt that "whether the new method can conserve the corresponding trends"

**To our knowledge, two approaches help preserve the physical consistency of climate model outputs and their trends: to apply trend-preserving multivariate methods or to use raw model outputs straightforwardly for impact analyses, accepting biases. Our proposed framework is based on calibrating a hydrologic model using of raw model outputs, assuming a consistent relative change (within climate simulations) from the reference to the future period. We acknowledged, in the manuscript, the requirement for calibration as a limitation of to the proposed framework, considering that it may disrupt the consistency of simulated hydrologic processes at the catchments scale. We also acknowledged that we do not know yet to which extent parametric compensation (resulting from calibration) affects the trends of the projected hydrologic responses, but we identified this issue as a key question for further**

research. However, our approach is based on the use of raw climate model outputs. We have verified that corresponding trends encrypted within climate model simulations are conserved. A comment has been added to the discussion [lines 432-438] to summarize the above arguments.

2.The method is still somewhat confusing, especially for the readers who are not familiar with the "asynchronous modelling" and the reason for "using an objective function that does not consider the temporal correlation". In addition, "let the hydrological model to run in an asynchronous fashion, considering that the same hydrologic model is expected to behave in a different way (the response to Reviewer #2)" also confused me. If the hydrological model behave differently in asynchronous fashions, the can the physical mechanism of the hydrological model be ensured? I appreciate the reference the author provide, but I still suggest to give a brief introduction to make it more readable.

The proposed asynchronous modelling framework has been previously explored by the authors. The two papers mainly focused on exploring different types of asynchronous objective-function and on the modelling framework that could be applied to implement more complex physically based description of hydrological processes, considering the scarcity meteorological fields such as air humidity, radiation, and wind speed. An asynchronous objective-function is equivalent to a signature-based calibration metric in the way it is designed. Both criterions aim to identify parametric solutions by optimizing the asynchronous statistical properties of the simulated hydrograph such as means (annual, seasonal or monthly), variance and quantiles, capturing the broad hydrologic behavior of a catchment instead of the precise sequence of hydrometeorological events observed at the catchment outlet. However, the purpose of asynchronous modelling is different from signature-based modelling. Based on the assumption that streamflow regime is a functional proxy of the corresponding forcing climate system, asynchronous modelling proposes a specific reconfiguration of the conventional hydroclimatic modelling chain, circumventing the (double, potentially redundant) requirement for meteorological observations typically used for post-processing raw climate model outputs and calibrating the hydrologic model.

Since climate models are not designed to simulate the observed sequence of meteorological events, we expect the resulting simulated hydrologic response to also be out-of-phase relative to streamflow observations. We thus expect the use correlation-based calibration metrics (such as NSE, KGE) to mislead the identification of sound and representative calibrated parameters within the asynchronous modelling framework. A paragraph has been added to the

**manuscript [lines 146- 153, see also 161- 163]. Finally, we believe it is acceptable for two parametric solutions, issued by two distinct modelling frameworks, to differ in their corresponding simulated hydrologic response.**

3.Another issue I am concerned is that, the comparison between the new and traditional methods (Figure 8 and Table 4). Here, the author said that the new method tend to have smaller uncertainties, but I note that the samples are different from the new and traditional approaches during the comparison. It is meaningful? In addition, some samples have large uncertainties in new method (e.g., crx4 in Site 4) but do not occur in traditional approach. Is this not the advantage of the traditional approach? Moreover, given that the new method shows clear weakness in representing the streamflow seasonal cycle compared to the traditional method, whether it is better in the future projection is doubtful. I do not think a lower uncertainties necessarily indicate a better projection especially when we do not have observations and its lower performance during historical period.

**We wish to clarify that we did not explicitly refer to the notion of uncertainty when describing and discussing Figure 8 and Table 4. We rather described the distributions of the projected change values, using the standard deviation to describe the spread of the corresponding distributions. We would argue that the spread affecting projected change values should theoretically include both uncertainties related the implementation of the modelling change and the naturally variability of the hydroclimatic system. We agree with Reviewer #3 that lower uncertainties do not necessarily translate into better projections. Also, we did not conduct a formal analysis comparing statistical distributions, we rather described how an impact modeler would interpret and communicate the projected change of the hydrologic regime. Our main intention was to determine if the corresponding conclusions would differ from a modelling framework to another. The proposed analysis is thus rated based on expert-based judgment rather than statistical significance.**

**Outlying change values are an obvious drawback of the proposed asynchronous modelling framework. We agree with Reviewer #3 (as discussed in the manuscript) that this drawback should be carefully taken into account using the asynchronous framework. We however demonstrated that the outlying change values are related to a poor representation of the annual hydrograph simulated over the reference period. In the manuscript, corresponding climate simulations have been formally excluded of the impact analysis, without affecting conclusions. For these reasons, we would not fully agree with Reviewer #3 that weaknesses in representing the streamflow seasonal cycle**

**affect the reliability of the projected change values issued by the asynchronous framework. Note that outlying change values can also be generated using the conventional approach (Appendix B).**

4.On the "quantile perturbation". Figure 10 shows the future projections based on quantile perturbation. I think it is reasonable using this method when the hydrological regimes remain stable in the future. However, the "tipping points" has been received much attention in recent years which is an important issue in future projection. Is the "quantile perturbation" suitable when the hydrological regimes changes? For example, a shift in seasonal cycle?

**We acknowledge that the quantile perturbation assumes a comparison between two stationary periods (reference vs future) and does not consider potential rupture in future trends. We believe shifts in the seasonal cycle could theoretically be more precisely assessed by applying sub-annual (monthly) perturbation factors. The point raised by Reviewer #3 highlights the fact that the asynchronous approach, as designed and presented in the manuscript, could rather be considered as an (hybrid-like) approach to assess vulnerability of water resource systems, instead of a pure top-down predictive assessment tool. Comments have been added to the discussion [lines 501-503 and 541-543].**